# Contact-free magnetic resonance imaging and spectroscopy with acoustic levitation

Smaragda-Maria Argyri ⓘ, Leo Svenningsson ⓘ, Feryal Guerroudj ⓘ, Diana Bernin ⓘ, Lars Evenäs ⓘ ✉ & Romain Bordes ⓘ ✉

Conventional magnetic resonance measurements often rely on the use of sample containers. This limits the implementation of time-resolved studies at the molecular level of liquid samples undergoing evaporation or other dynamic phenomena that require access to the liquid-gas interface. In this study, we developed a demagnetized acoustic levitator to perform magnetic resonance studies on liquid samples, in a contact-free manner. The performance of the levitator inside a 7.05 T magnetic field was examined and magnetic resonance images of the levitator and the levitated samples were acquired. Then, we collected magnetic resonance spectra of the levitated droplets by applying localized and non-localized pulse sequences and we examined the effect of the droplet shape on the chemical shift. Additionally, we conducted time-resolved experiments on pure solvents and mixtures, and captured physical and chemical molecular interactions, in real-time. This approach enables contact-free studies at the molecular level of dynamic phenomena on a microliter droplet using magnetic resonance techniques.

Previous attempts to study self-standing droplets in a magnetic field have predominantly focused on buoyancy forces of immiscible liquids[1–3], or molten aluminum beads maintained aloft by aerodynamic levitation[4–7]. However, these methods do not allow direct access to the interface and are restricted to non-volatile samples. Recent advancements have explored magnetic resonance studies on surface-resting sessile droplets[8,9], providing real-time dynamic insights, yet, without granting a fully contact-free environment. Acoustic levitation enables the implementation of contact-free studies on small-volume objects (0.5−5 μL). This is achieved by generating a stable acoustic pressure field with the use of ultrasonic waves that are set up to create a standing wave ($f \geq 20$ kHz)[10–12]. As a result of the standing wave, the acoustic pressure field has areas of low and high pressure, known as nodes and antinodes, respectively. Levitation occurs at the nodes. The pivotal advantage of acoustic levitation, compared to other levitation techniques, is that the sample does not need to present specific properties (e.g., magnetic or dielectric) to undergo levitation. Furthermore, the shape of liquid droplets can be controlled based on the surface tension, volume, and the applied acoustic radiation force[13–15]. This approach has recently been combined with various analytical techniques, demonstrating its applicability and importance in contact-free studies. For instance, acoustic levitation has been successfully coupled with small angle X-ray scattering for materials characterization[16,17]. In the field of molecular biology, Raman spectroscopy was coupled with acoustic levitation for the study of cells[18,19] and photochemical reactions[20], providing valuable insights into cellular behavior and molecular interactions. Protein crystallization studies have also benefited from its combination with X-ray diffraction techniques[21].

Recent advances in acoustic levitation instrumentation have significantly improved the performance of phased-array acoustic levitators by properly adjusting the design parameters[22]. Specifically, it has been shown that by arranging transducers in a hexagonal configuration and properly adjusting the distance between the opposing arrays, high acoustic levitation force, and micrometer control on the displacement of the levitated sample were accomplished. A challenge in combining acoustic levitation and magnetic resonance studies is the effect of the magnetic field on the piezoelectric components of the

Department of Chemistry and Chemical Engineering, Chalmers University of Technology, Gothenburg, Sweden. ✉e-mail: lars.evenas@chalmers.se; bordes@chalmers.se

ultrasonic transducers, and on the overall performance of the acoustic levitator. Other practical considerations are the lack of visual access to the sample inside the magnet and the need to address the presence of magnetic elements in the components of the levitators.

In this study, we used the design of the most performing acoustic levitator, derived from one of our previous studies, which we initially engineered with a primary emphasis on levitation capacity and stability[22]. A mechanical lift incorporated the acoustic levitator and allowed the transportation of the sample at the center of the magnet bore. The operating ability of the levitator in the 7.05 T (300 MHz) magnetic field was evaluated by recording the electric current at different operating frequencies of the transducers. We acquired magnetic resonance (MR) images of an acoustically levitated hexadecane droplet using three different radio frequency pulse sequences (fast low angle shot magnetic resonance imaging (FLASH[23]), true fast imaging with steady-state free precession (True-FISP[24]), and rapid acquisition with relaxation enhancement (RARE[25]) which acted as direct proof of the successful acoustic levitation of the sample inside the magnet. Moreover, the evaporation of a water droplet was monitored and quantified through MR imaging. Following, we recorded MR spectra by applying localized and non-localized pulse sequences, and we investigated the effect of the acoustic force applied on a hexadecane droplet, on the MR spectra. Lastly, we levitated a 50 wt% aqueous triethylene glycol (TEG) droplet in the magnet, we collected MR images and spectra during drying, and followed the image and spectral evolution, over time. Overall, this study demonstrates the implementation of contact-free MR studies through acoustic levitation and showcases numerous real-time, dynamic studies on single, self-standing droplets. Dynamic processes and phenomena can be studied on a molecular level without the presence of a container.

## Results and discussion

### Acoustic levitation

The design of the acoustic levitator was optimized[22], and consisted of $18 \times 2$ demagnetized, ultrasonic transducers, positioned in a hexagonal arrangement. The levitator presented high acoustic levitation forces and micrometer-level stability over the levitated sample,[22] while its compact size allowed the insertion in the 66 mm diameter [1]H MR imaging probe. Furthermore, we built a mechanical lift (Supplementary Fig. S1) which transported the acoustic levitator and the levitating sample, inside the bore of the magnet. Following, we examined the electrical response of the demagnetized acoustic levitator in the 7.05 T magnetic field by varying the operating frequency of the transducers and measuring the current consumption, at three different voltages (7 V, 10 V, 12 V), when no sample was being levitated. That way, we evaluated the frequency response close to the resonance frequency of the transducers, at 40 kHz, inside and outside the magnetic field (Supplementary Fig. S2). Due to eddy currents, the frequency response profiles became broader and the current consumption decreased inside the magnet. The reduced current flow through the device was expected to reduce the performance of the levitator inside the magnet. For that reason, we operated the device at 0.5 to 1.0 V higher than the voltage we would normally use for studies outside the magnet. In a recent publication[22], we showed that the performance of the levitator can be improved by fine-tuning the operating frequency of the transducers to match the maximum current consumption. It was shown that as long as the operating frequency remained within ±1 kHz of the resonance frequency of the transducers (40 kHz), up to 10% improvement was achieved in terms of droplet stability and levitation strength. However, in this case, the frequency response profiles inside the magnet are broader, and the current consumption peaks did not shift significantly from the resonant frequency of the transducers at 40 kHz at higher voltage values, thus an operating frequency of 40 kHz was selected throughout the study.

The effect of the temperature of the medium on the acoustic pressure generated by the acoustic levitator was also investigated through simulations (Supplementary Fig. S3). This aspect was examined due to a potential increase in temperature resulting from strong localized MR pulse sequences, which could affect the density and speed of sound of the medium (i.e., air). Experimentally a temperature increase of the medium by 1–2 °C was observed during the measurements. For that reason, we simulated and compared the acoustic pressure along the $x$-, $y$-, and $z$-axes at 20 °C, 25 °C, and 30 °C. It was found that the acoustic pressure did not change significantly close to the central node. Consequently, within the limits of the simulations, the temperature increase of the medium was considered negligible on the operation of the levitator during the MR data acquisition.

Another phenomenon related to acoustic levitation that may influence magnetic resonance measurements is acoustic streaming. This phenomenon refers to the presence of a secondary acoustic force, leading to a stream of air around the levitating droplet, that may lead to a flow of fluid inside the droplet[26]. This phenomenon has previously been reported in studies with the so-called Langevin horn, which is an acoustic levitator consisting of a single, high-power transducer and a reflector. The Langevin horn generates a simple acoustic pressure field, similar to a standing wave, with extremely low lateral forces[15,22]. In the case of the acoustic levitator used in this study, a more complex acoustic pressure field is generated[22], which may affect the internal forces in the droplet differently. A previous study on drying conducted with a multiple-transducers acoustic levitator showed that sodium chloride (NaCl) crystals sedimented inside the levitating droplet; consequently, no internal flow was present[27]. This was supported in a recent publication, where aluminum oxide and silica particles with different sizes sedimented inside a water droplet within a few minutes[28]. This also indicates that adding two miscible droplets in the acoustic node will lead to the mixing of the droplets due to diffusion and not due to potential external effects related to the acoustic levitator. However, we have observed experimentally that the droplet can rotate around the $z$-axis in a random fashion. This can lead to motion artifacts in MR images, while it could also hinder diffusion measurements. Yet, recent studies have shown that it is possible to control the rotation of non-spherical[29] and spherical objects[30], which broadens the possibilities of contact-free studies with acoustic levitation.

### Magnetic resonance imaging on self-standing droplets

The acoustic levitator was operated at 9.0 V and a liquid droplet of hexadecane was deposited at the central node using a glass syringe with a metallic needle. Then, the acoustic levitator was transported to the detection zone of the MRI probe using the mechanical lift (Supplementary Fig. S1). The levitated droplet was visualized with the gradient-echo-based sequence FLASH (Fig. 1a), the True-FISP (Fig. 1b), and the spin-echo-based sequence RARE (Fig. 1c), with an in-plane resolution of 312.5 μm/pixel, across the axial ($xy$), sagittal ($xz$), and coronal ($yz$) slices, each with a thickness of 1 mm. The FLASH sequence presented the lowest input energy at 1.4 W, followed by True-FISP at 12.6 W, and RARE at 41 W and 324 W, for the excitation and refocusing pulse, respectively. The high input energy from RARE at this in-plane resolution caused the levitated droplet (Supplementary Fig. S8) to burst during the measurements. The MRI shown in Fig. 1c, along the $xz$ plane, is the only slice in which the droplet was visible. In Supplementary Fig. S9, fragments of the droplet are visible in the different slices, along the sagittal ($xz$) plane, while no signal was recorded along the axial ($xy$) and coronal ($yz$) planes. However, by choosing a hyperbolic secant (sech) function for the excitation pulse and a cubic sinc (sinc3) function for the refocusing pulse, we minimized the input power of the pulses of RARE to 8.75 W and 114.24 W, respectively. Furthermore, by decreasing the in-plane resolution to 1 mm/pixel, we acquired MR images of the droplet along the three planes, where parts of the transducers were also visible (Supplementary Figs. S4-S6).

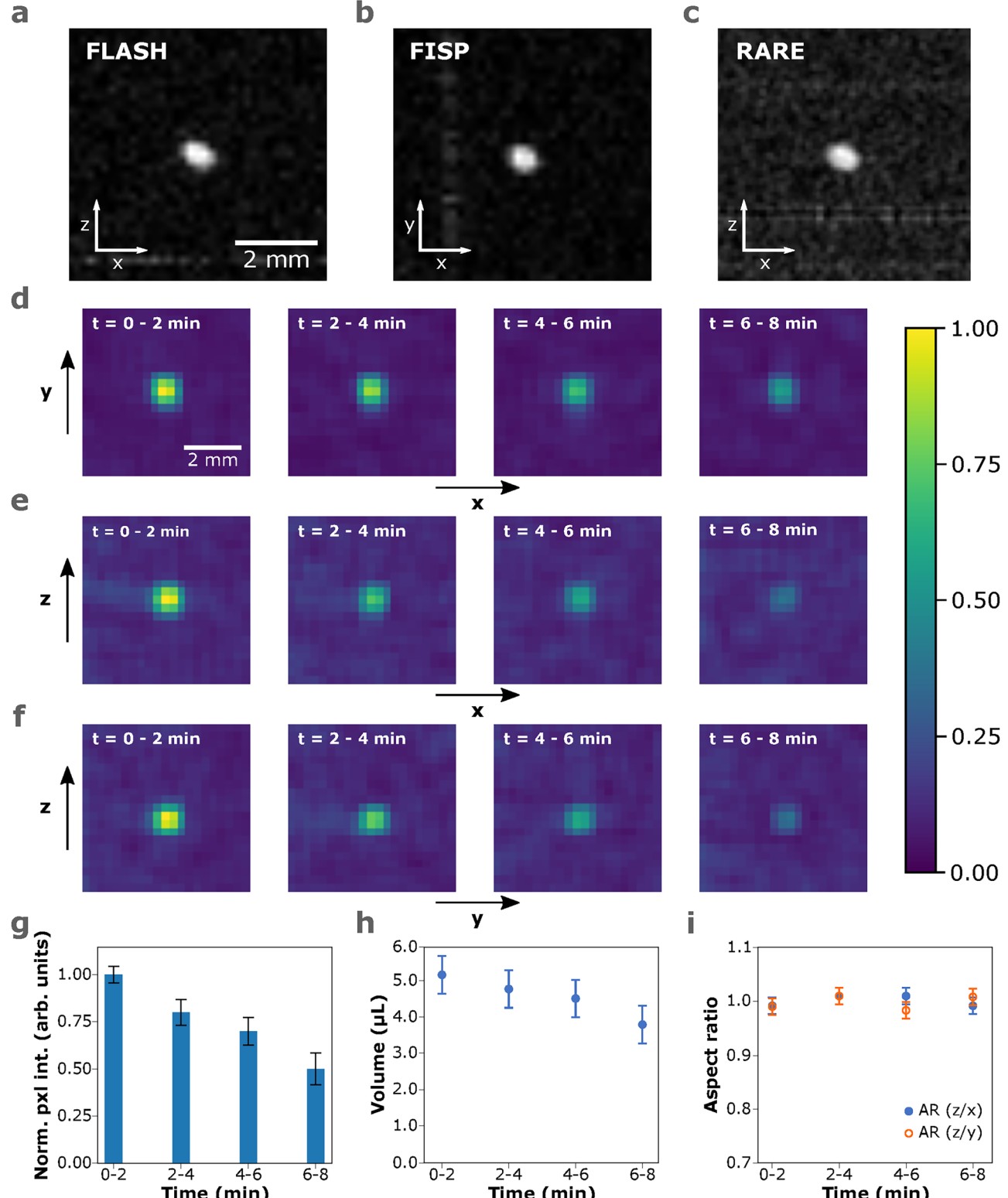

**Fig. 1 | Magnetic resonance imaging of microliter, self-standing droplets in real-time.** MR images of an acoustically levitated droplet of hexadecane (-1.34 μL) acquired with (**a**) FLASH, **b** True-FISP, and (**c**) RARE. MR images of an acoustically levitated droplet of water, with an initial volume of ~5.15 μ, over a period of 8.5 min, were acquired with FLASH along the (**d**) axial (*xy*), **e** sagittal (*xz*), and (**f**) coronal (*yz*) planes. The color bar expresses the normalized pixel intensity of the (**d**–**f**) subfigures. The time in the inset refers to the period of acquisition. **g** Average normalized maximum pixel intensity over time. Calculated (**h**) volume, and (**i**) aspect ratio of the water droplet from the FLASH MR images, over time.

Following, we levitated a water droplet at 10.0 V, and recorded four consecutive MR images with FLASH over 8.5 min, where each repetition lasted for approximately 2.1 min. In Fig. 1d–f, it is evident that the pixel intensity decreased, on all planes, as the evaporation took place. From the MR images, we plotted the pixel intensity around the droplet with respect to the axial distance (Supplementary Fig. S11) and determined the width at half the height, which was used as an estimation of the diameter of the droplet (Supplementary Table S1). From those values, we draw overlays on the MR images of the droplet (Supplementary Fig. S10), and we determined the volume from Supplementary Eq. (S1). The values determined were in good agreement with previous work on similar levitators[15,17,22]. From the normalized, maximum pixel intensity on each plane, we calculated the average pixel intensity of each repetition (Fig. 1g), and from the droplet dimensions we determined the volume (Fig. 1h) and aspect ratio (Fig. 1i) of the droplet, over time. The change in pixel intensity relates to the number of protons at a certain period of time, which, in this case, relates to the volume of water. By comparing the pixel intensity with the volume estimations, a loss of material due to evaporation was observed and the general trend supported the expected decrease. However, the volume estimations had higher values, potentially due to the in-plane resolution. MR images with higher in-plane resolution (250 μm/pixel, i.e., 25 times larger than the spatial stability of the levitator) were recorded with the pulse sequence True-FISP (Supplementary Fig. S12); though, artifacts were observed on the xz and yz planes, potentially due to higher sensitivity of the pulse sequence to the displacement of the droplet.

The aspect ratio of the droplet relates to the sphericity of the droplet, where an aspect ratio of 1 indicates a spherical droplet, and a lower aspect ratio indicates an oblate[15]. In Fig. 1i, it is observed that the aspect ratio is consistently close to 1, due to the low acoustic radiation forces applied on the droplet, at 10 V inside the magnet. Furthermore, it is observed that the aspect ratios calculated from the xz and yz planes differ by less than 3%, due to the x and y dimensions of the droplet being approximately equal. These measurements showcase the possibility of visually following dynamic phenomena, such as solvent evaporation, and characterizing the material in real time.

**Magnetic resonance spectroscopy on self-standing droplets**

In Fig. 2a, we present the MR spectra that was collected by applying pulse sequences within a defined voxel (i.e., localized spectroscopy), and within the whole detection volume of the probe (i.e., non-localized spectroscopy), while a droplet of canola oil that was acoustically levitated at 9.0 V. Those spectra were compared to a standard NMR spectrum acquired using a 5 mm probe in a 7.05 T magnet. The localized pulse sequences STEAM (stimulated echo acquisition mode[31]), PRESS (point resolved spectroscopy[32]), and ISIS (image-selected in vivo spectroscopy[33]) were adjusted to improve the signal-to-noise ratio. In all cases, we acquired spectra along a $3 \times 3 \times 3$ mm voxel, using the bp32 pulse shape for all three excitation pulses, which led to minimum echo time. For STEAM and PRESS, it was found that applying minimum gradient spoiler strength and duration improved the spectral acquisition, while in the case of STEAM, the mixing time (i.e., the time interval between the second and third radio frequency pulses[31]) was minimized, too. In the case of spin-echo-based sequences, gradient spoilers serve as post-acquisition gradient pulses aimed at mitigating the persistence of residual transverse magnetization. This residual magnetization can arise from incomplete dephasing of nuclear spins during signal acquisition, potentially leading to undesired coherent signal reconstructions or artifacts in the resulting images or spectra. By applying gradient spoilers following signal acquisition, one can effectively dephase any remaining transverse magnetization, ensuring that the acquired signal reflects the true magnetization state of the sample. However, if the strength or duration is too strong, the gradient spoiler can dephase not only the remaining transverse magnetization but also

the desired signal, leading to a significant signal loss and artifacts. The choice of optimum mixing time in STEAM relates to the relaxation properties of the sample. A longer mixing time allows for more extensive magnetization transfer and J-coupling interactions, potentially enhancing the visibility of coupled spins and improving spectral resolution. However, excessively long mixing times can lead to excessive signal loss due to relaxation effects and may not necessarily provide additional benefits in terms of spectral quality. For the non-localized pulse sequences, NSPECT (Non-Localized Spectroscopy, Bruker) and Single Pulse, we set the same repetition time and the number of averages for both of them and no further adjustments were made. It was observed that STEAM resulted in broader line widths, followed by PRESS, and ISIS. The non-localized pulse sequences led to more distinct signals in comparison to the localized ones; however highly non-linear baselines were recorded (Supplementary Fig. S15), which was adjusted by applying splines baseline correction. In comparison to the standard NMR spectrum of canola oil, we observed that the ISIS and the non-localized pulse sequences led to equally well-resolved spectra, as all the main signals are distinguished, with similar line widths (Supplementary Tables S3 and S4) which indicates field homogeneity inside the droplet.

The canola oil spectra were calibrated by using the highest signal in intensity at 1.51 ppm, as a reference. In practice, however, the shape of the droplet affects the local magnetic field strength[34], which results in a frequency shift. In the Supplementary Information, we determined the theoretical frequency shift of hexadecane, depending on the shape of the droplet ranging from a sphere (i.e., aspect ratio = 1), to an oblate spheroid (i.e., aspect ratio < 1) from Supplementary Eq. (S2–S16)[35]. This change in resonance frequency, results in a chemical shift according to Supplementary Eq. (S17). For simplicity, we set the chemical shift of a spherical droplet at 0 ppm difference from the chemical shift that would appear in case tetramethylsilane (TMS) was used as a reference compound (Supplementary Eq. (S18)). In Fig. 2b, we calculated this chemical shift difference for the case of hexadecane, in a 300 MHz magnet. It is observed that the chemical shift increased as the droplet shape changes from a sphere to an oblate.

This effect was investigated by levitating a droplet of hexadecane and collecting a series of MR spectra at different voltages with the localized pulse sequence ISIS (Supplementary Fig. S16-S17). The voltage was used as a means to control the acoustic pressure applied on the droplet and thus change the shape from spherical at low voltage, to oblate at higher voltage, as described in previous studies[13–15]. In Fig. 2c, representative examples of the repeated measurements are shown. It is observed that the chemical shift increased as the voltage increased, which is in agreement with the theory presented in the Supplementary. The shape of the droplet was inspected with the True-FISP MR imaging pulse sequence (Supplementary Fig. S13-S14), with an in-plane resolution of 250 μm/pixel, and it was found that at 9.0 V the aspect ratio was close to 1.0 (Supplementary Table S2). Hence, the chemical shift of the spectra at 9.0 V was calibrated by setting the methylene signal ($CH_2$) at 1.26 ppm[36]. Then, we determined the theoretical aspect ratio change based on the chemical shift difference from the chemical shift of 1.26 ppm as the voltage varied. We found a continuous increase in chemical shift from 1.26 for 9.0 V to 1.55 ppm for 12.0 V. Based on the theory, these chemical shift changes correspond to aspect ratios from 1.00 to 0.88. Furthermore, it is observed that the chemical shift difference between the two signals of hexadecane is $0.4 \pm 0.002$ ppm on each spectrum, regardless of the voltage (Supplementary Fig. S18). Following, we compared the theoretical aspect ratios with the MR images at 9.0, 10.0, 11.0, and 12.0 V (Supplementary Fig. S13-S14). In Fig. 2d, it is observed that the aspect ratio values determined by combining the chemical shift of the MR spectra and the theory are in good agreement with the values determined from the MR images, underscoring that the difference in chemical shift gives a good estimation of the shape of the levitated droplet. Furthermore, our findings

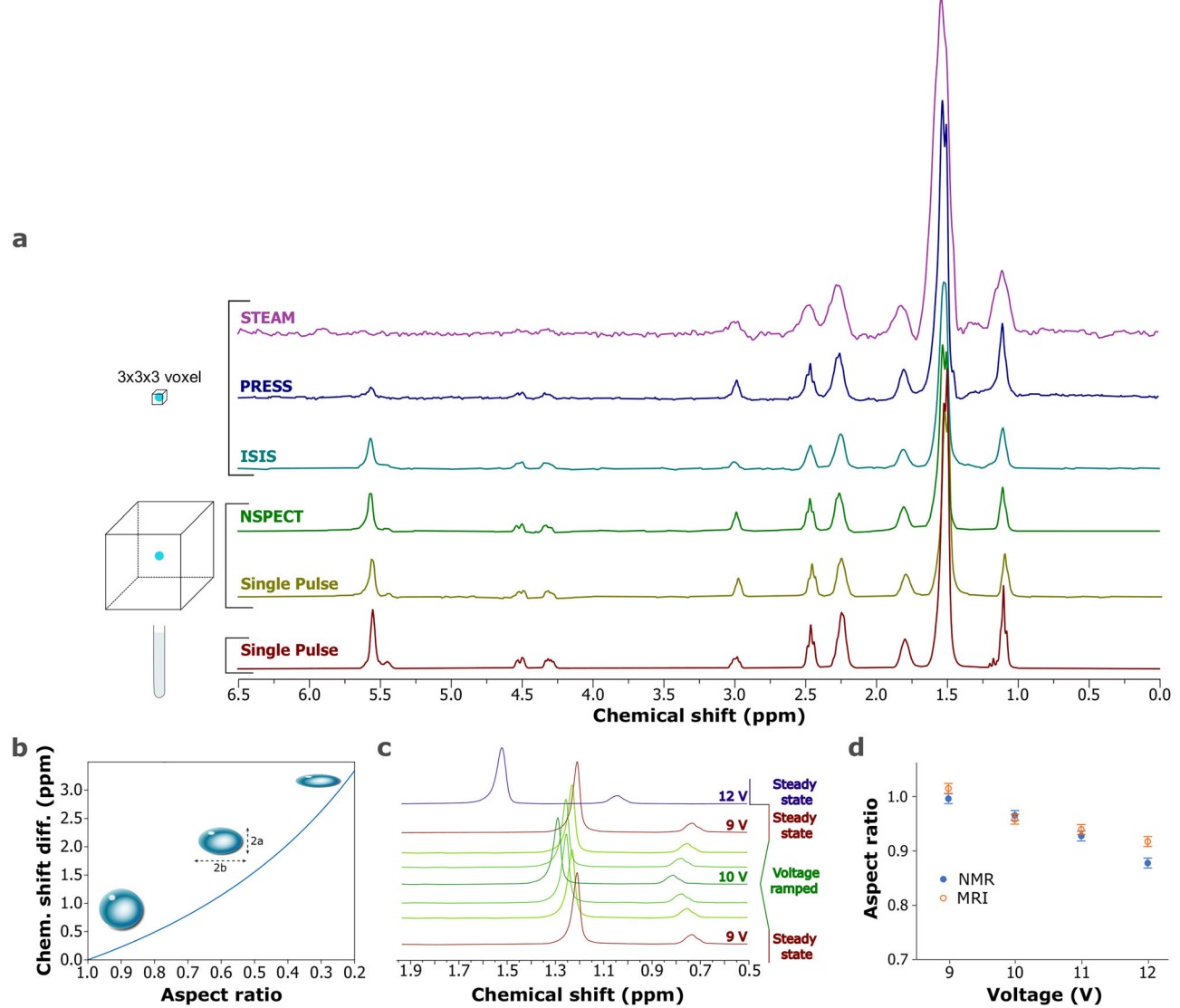

**Fig. 2 | Magnetic resonance spectroscopy on microliter droplets and effect of droplet shape. a** MR spectra of a canola oil droplet, levitated at 9.0 V, acquired in a 7.05 T magnet, with a 66 mm MR imaging probe, by applying the localized pulse sequences STEAM, PRESS, and ISIS on a 3 × 3 × 3 mm voxel, and the non-localized pulse sequences NSPECT, and Single Pulse. Bottom spectrum: Standard NMR spectrum of canola oil acquired in a 7.05 T magnet with a 5 mm NMR probe. The chemical shifts of the spectra acquired on the levitated droplet have been shifted to match the spectrum of canola oil in the glass tube. **b** Theoretical chemical shift difference for a hexadecane droplet with respect to the aspect ratio. From left to right: illustrations of droplets with aspect ratios of 0.9, 0.6, and 0.3. **c** Representative examples from a series of ISIS MR spectra of an acoustically levitated hexadecane droplet (-3.75 μL) at different voltages (i.e., aspect ratio). **d** Aspect ratios of hexadecane determined by combining the MR spectra with the theory presented in the Methods, compared to the aspect ratio determined by MR images of hexadecane levitated at 9 V, 10 V, 11 V, and 12 V.

suggest that the employed pulse sequences did not induce a notable increase in the temperature of the droplet, since if that was the case a change in chemical shift would be recorded over time, for constant voltage (Supplementary Fig. S17).

### Probing molecular interactions within self-standing droplets

Subsequently, we studied the evaporation of a 50 wt% aqueous TEG droplet by MR imaging and spectroscopy at 9.0 V. TEG was chosen for its high water affinity, leading to strong interactions through hydrogen bonding[37]. In Fig. 3a–c the FLASH MR images of the droplet are shown along the sagittal (*xz*), coronal (*yz*), and axial (*xy*) planes, over a period of 15 min. In Fig. 3d, the pixel intensity decreased by 27 ± 5% within the first 3–6 min, due to the water evaporation, and reached a plateau during the next 6–15 min. In Fig. 3e, a series of ISIS MR spectra of a 50 wt% TEG droplet, levitated at 9.0 V, over a period of 16 min are depicted. The methylene ($CH_2$) signal was calibrated at 3.67 ppm, in

accordance with ref. 38, and the -OH signal from water and TEG-OH appeared at approximately 4.67 ppm. Initially, the water content was high, consequently, the -OH signal was mainly dominated by the water contribution. As evaporation took place, the solution reached an intermediate state where there was a more equal -OH exchange between water and TEG due to hydrogen bonding interactions, resulting in a broadened signal between the chemical shift positions of each constituent. Finally, the water evaporated to the extent that there was a major abundance of TEG-OH protons, thus leading to the emergence of the TEG-OH signal at 4.45 ppm. In Fig. 3f, we plotted the normalized integrated area under the TEG signals between 3.20 and 4.20 ppm and the peak from the -OH protons, within the range of 4.20 and 5.50 ppm. The integral that corresponds to the -OH protons reached a plateau within the first 5–6 min, which is in agreement with the MR imaging results. Furthermore, the final, equilibrium ratio, after 15 min, between the normalized TEG and TEG-OH integrals was 6:1,

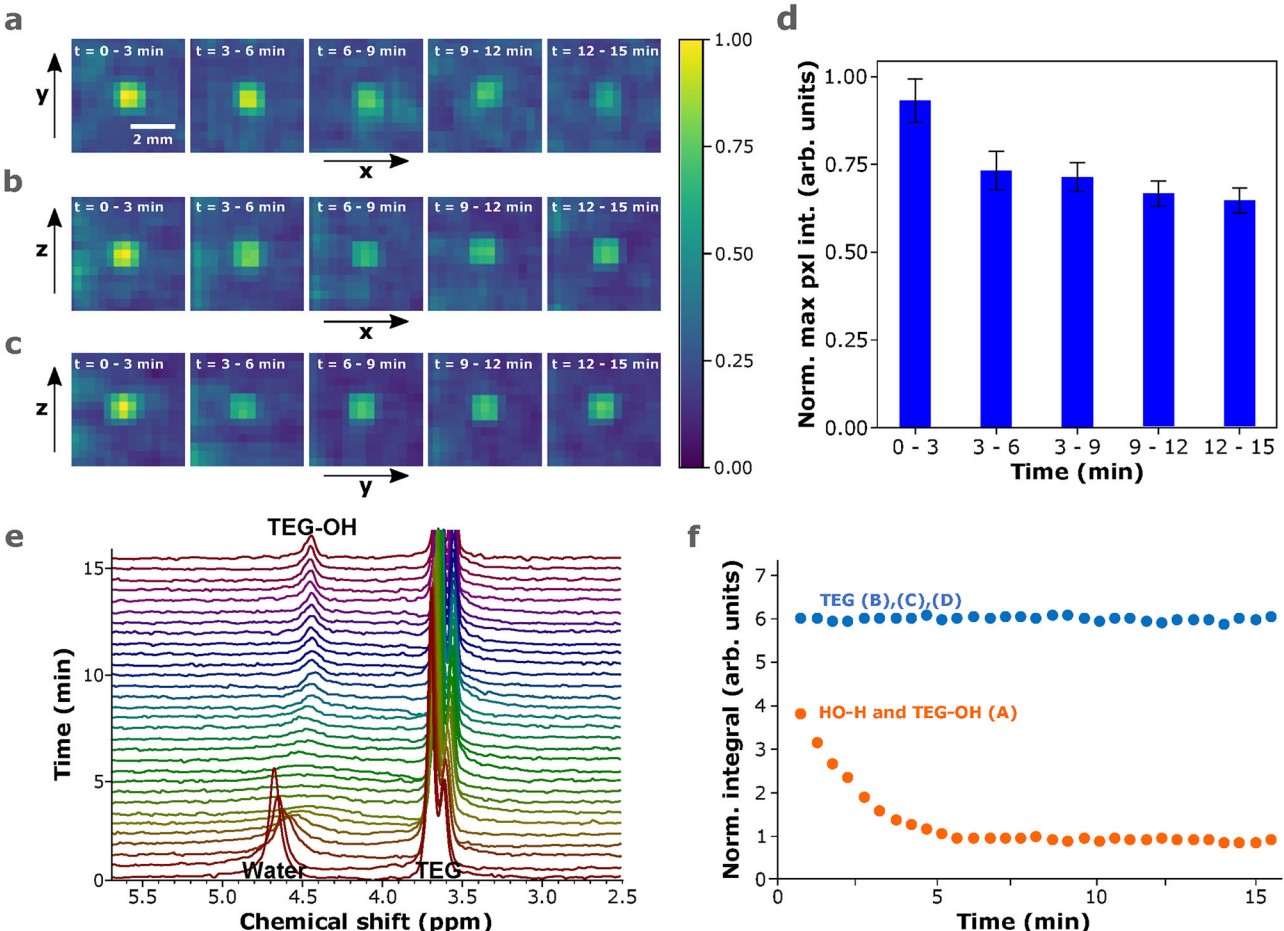

**Fig. 3 | Probing molecular interactions and quantification within microliter droplets.** FLASH MR images of a ~4 μL, 50 wt% TEG aqueous droplet, levitated at 10.0 V along the (**a**) axial (*xy*), **b** sagittal (*xz*), and (**c**) coronal *yz* planes, over a period of 15 min, **d** normalized maximum pixel intensity with respect to time,

**e** Time-resolved ISIS MR spectra of a ~4 μL, 50 wt% TEG aqueous droplet, levitated at 10.0 V, over a period of approximately 15 min, and (**f**) normalized integrals of MRS signals over time.

which corresponds to the ratio of hydrogen atoms present in the main chain and the carbons located at the extremities of the chain, thus indicating that water had completely evaporated. This showcases the accuracy and reliability of the measurements on the microliter droplet. Moreover, we followed the chemical shift of the signals over time (Supplementary Fig. S19), and found that the TEG signals shifted upfield by approximately 0.05 ppm. This indicates that the droplet became slightly more spherical as the water evaporated, due to a decrease in volume combined with a minor decrease in surface tension, under constant acoustic pressure, which in is line with previous studies[15]. These measurements support that molecular processes taking place during evaporation can be monitored in real-time, in an acoustically levitating droplet within a high magnetic field environment.

## Outlook and summary
Herein, we coupled acoustic levitation with a 7.05 T magnet and performed magnetic resonance studies on microliter liquid droplets i.e., 0.5–5 μL, on aqueous systems. While previous studies have explored single, self-standing droplets using techniques such as aerodynamic levitation[4–7] and control of immiscible liquids through fluid flow[1–3], we performed magnetic resonance studies on microliter liquid droplets exclusively surrounded by air. This allowed the acquisition of MR images and high-resolution spectra of microliter, self-standing droplets, by using an acoustic levitator that was optimized for integration in a high-field magnet. The non-localized MR spectroscopy pulse

sequences yielded the highest spectral resolution, equal or superior to that of the standard NMR tube spectrum collected in a 300 MHz magnet with a 5 mm NMR probe. We varied the acoustic pressure exerted on the droplet and observed a downfield shift as the aspect ratio decreased. This direct correlation between the shape of the droplet and the chemical shift was supported by theoretical analysis, which showed that the local magnetic field strength in the droplet changes depending on the shape. Lastly, we monitored the real-time drying process of a 50 wt% TEG droplet by imaging and spectroscopy and captured the -OH proton exchange dynamics during water evaporation. These findings are instrumental for materials characterization and real-time investigation of dynamic phenomena, such as chemical and enzymatic reactions at interfaces, or protein crystallization. Further development of the methodology including other aspects of MR imaging and spectroscopy can be anticipated.

## Methods
### Materials
Hexadecane and triethylene glycol (TEG) were obtained from Sigma-Aldrich Sweden AB. Canola oil was acquired from the local store. The 50 wt% TEG aqueous solution was prepared in Milli-Q water (resistivity of 18.2 MΩ · cm, at 25 °C, Merck Group, Sweden).

### Acoustic levitator
The acoustic levitator was connected to a power supply (RND 320-KD3005P DC, RND Electronics, Switzerland), through which the

operating voltage was controlled. The operating frequency of the transducer was set, through an Arduino Uno board and a module Si5351, at 40 kHz. The driving signal was amplified with an L298N H-bridge. A $\pi$ rad phase difference was applied between the two transducer halves. The design and construction of the acoustic levitator Mk3 are presented in ref. 22, and further details can be found on https://git.chalmers.se/bordes/OmniLev. This levitator enabled the levitation of liquid droplets in the volume range of 0.5–5 µL and a density range of 0.8 to 1.2 g/cm$^3$. Yet, materials with higher densities have been tested with similar levitators in previous studies[22,39]. For this study, a demagnetized version was used, for which the magnetic pins of the ultrasonic transducers (Manorshi, MSO-P1040H07T) were replaced with non-magnetic ones, through soldering, before attaching them to the 3D-printed scaffold. The hexadecane droplets were deposited through a 10 µL Hamilton glass syringe with a metallic needle attached to it. The rest of the droplets were deposited through a disposable 2 ml plastic syringe with a disposable metallic needle attached to it.

### Frequency response

The circuit design and control of the operating frequency of the transducers is described in ref. 22. In this work, we varied the operating frequency of the transducers between 36 kHz and 44 kHz and recorded the current consumption of the acoustic levitator using a module ACS70331, when the device was operating at 7 V, 10 V, and 12 V. All measurements were repeated three times.

### Mechanical lift

The mechanical lift (Supplementary Fig. S1) consisted of 3 carbon fiber tubes with a length of 60 cm, passing through 4 circular disks with an outer diameter of 67 mm, and an inner diameter of 57 mm. The acoustic levitator was placed inside a 3D-printed holder that allowed the transportation of the device inside the MRI bore. It was experimentally observed that the tuning and matching of the receiver frequency were affected by the position of the cables inside the magnet. For that reason, the cables were attached with zip ties to a 10 cm 3D printed, plastic rod, attached at the bottom of the 3D-printed holder.

### Magnetic resonance imaging and spectroscopy

We used a super wide bore (89 mm) Bruker Avance III 300 MHz $^1$H (7.05 T) equipped with a 66 mm probe, set in a static gradient system for imaging. We performed shimming on a 5 mm diameter glass tube containing the sample of interest. The mechanical lift with the acoustic levitator was positioned at the detection zone of the MRI probe. Manual tuning and matching were performed while the acoustic levitator was operating, with and without a levitated droplet. Then, we used the FLASH pulse sequence with a field of view of 64 × 64 mm to locate the droplet. (Supplementary Fig. S7).

The magnetic resonance images of hexadecane, shown in Fig. 1a–c, were recorded with the pulse sequences FLASH, True-FISP, and RARE. For the pulse sequence FLASH, we used a 20 × 20 mm field of view, 64 × 64 pixel region of interest (ROI), 1 mm slice thickness, repetition time of 1500 ms, and the bp32 excitation pulse shape. For True-FISP we used the same field of view, ROI, and slice thickness, 5 averages, echo time of 1.143 ms, repetition time of 2.287 ms, 21 segments, segment mode sequential, scan repetition time of 35 ms, and a 90° pulse. For RARE we used the same field of view and ROI, 8 averages, repetition time of 3000 ms, echo time of 2.28 ms, RARE factor 1, and the bp32 excitation and bp32 refocusing pulse. In all cases, the scan time was approximately 6.5 min and the images were depicted using the Bruker software.

The magnetic resonance images of Milli-Q water droplets, shown in Fig. 1d–f, were recorded with FLASH and True-FISP pulse sequences. For the pulse sequence FLASH, we used 45 × 45 mm field of view, 64 × 64 pixel ROI, 1 mm slices thickness, 1 average, echo time of

1.84 ms, repetition time of 2000 ms, and we applied 4 repetitions, with a total scan time of 8 min and 32 sec. The images were plotted with Python and adjusted by applying a 3 × 3 kernel with values of 0.1. For images collected with the pulse sequence True-FISP, shown in Supplementary Fig. S12, we used 32 × 32 mm field of view, 128 × 128 pixel ROI, 1 mm slices thickness, 1 average, echo time of 1.783 ms, repetition time of 3.567 ms, 64 segments, 32 slices, bp32 excitation pulse shape, and applied 4 repetitions with a total scan time of 16 min and 25 sec. The images were plotted with Python.

The magnetic resonance images of hexadecane at 9 V, 10 V, 11 V, and 12 V, shown in Supplementary Fig. S13, were recorded through the FLASH pulse sequence. We set the field of view at 40 × 40 mm, ROI at 64 × 64 pixel, 1 mm slices thickness, 1.84 ms for echo time, and 1000 ms for repetition time (scan time: 2 min and 36 sec). The images were plotted with Python and adjusted by applying a 3 × 3 kernel with values of 0.1.

The MR spectra of canola oil, shown in Fig. 2a, were recorded with the pulse sequences STEAM, PRESS, ISIS, NSPECT, and Single Pulse. Before initiating the localized pulse sequences (STEAM, PRESS, and ISIS) the droplet was located by imaging, with the pulse sequence FLASH, and then, we selected a 3 × 3 × 3 mm voxel around it (Supplementary Fig. S7). For the pulse sequence STEAM, we used 200 averages, 1500 ms repetition time, 3.053 ms echo time, bp32 excitation and bp32 refocusing pulses, minimum gradient spoiler strength, and mixing time (scan time: 5 min and 0 sec). For PRESS, we used 100 averages, 1500 ms repetition time, 7.967 ms echo time, bp32 excitation and bp32 refocusing pulses, and minimum gradient spoiler strength and duration (scan time: 2 min and 30 sec). These values were minimized by using the integrated feature of the software of the instrument and no further modification was conducted. For ISIS, we used 8 averages (64 ISIS averages), 1500 ms repetition time, and bp32 excitation pulse (scan time: 1 min and 36 sec). We applied a 2 Hz exponential line broadening, the phase was manually corrected, and the baselines were corrected by applying a Bernstein polynomial fit with an order of 3. The non-localized MR spectra of canola oil were recorded with NSPECT and Single Pulse by applying 50 averages and 1500 ms repetition time in both cases without further adjustments (scan time in both cases: 1 min and 15 sec). We applied a 2 Hz of exponential line broadening, the phase was manually corrected, and the baselines of the spectra were corrected by applying splines. The NMR spectrum of canola oil in the glass tube was performed at KTH Royal Institute of Technology, Stockholm, in a 300 MHz Bruker magnet with a 5 mm probe, by applying a 90° Single Pulse with 16 averages and 1500 ms repetition time. We applied 2 Hz exponential line broadening, the phase was manually corrected, and the baselines were corrected by applying a Bernstein polynomial fit with an order of 3.

The steady-state MR spectra of hexadecane at 9 V, 10 V, 11 V, and 12 V (Supplementary Fig. S16) were recorded with an ISIS pulse sequence. Prior to that, the droplet was located with a FLASH pulse sequence. We set the field of view at 40 × 40 mm, ROI at 64 × 64 pixel, 1 mm slices thickness, 1.84 ms for echo time, and 1000 ms for repetition time (scan time: 1 min and 36 sec). To collect the ISIS MR spectra, we applied 8 averages (64 ISIS averages), 1500 ms repetition time, on a 3 × 3 × 3 mm voxel (scan time: 1 min and 36 sec). Then, we applied 5 Hz exponential line broadening, Bernstein polynomial fit with an order of 3 baseline correction, and manual phase correction. The series of MR spectra on a hexadecane droplet (Supplementary Fig. S17) were recorded with the ISIS pulse sequence by applying 1 average (8 ISIS averages), 3000 ms repetition time, on a 3 × 3 × 3 mm voxel (scan time of each spectrum: 24 sec). Initially, a series of 15 spectra were recorded with a driving voltage of 9 V. Then the voltage was varied by 0.1 V with a 30 sec delay before each measurement. The second period of 9 V steady-state measurements was gathered continuously followed by an approximate 2 min delay, and succeeded by continuous measurements at 12 V under steady-state. We applied 5 Hz exponential line

broadening, the phase was manually corrected, and the baselines were corrected by applying Bernstein polynomial fit with an order of 3.

The MR images of the 50 wt% TEG-water droplet, that was levitating at 9 V, shown in Fig. 3a–c, were recorded with a FLASH pulse sequence. We used 5 repetitions, with 45 × 45 mm field of view, 64 × 64 pixel ROI, 1 mm slice thickness, 1 average, 1.84 ms for echo time, and 2500 ms for repetition time. The total scan time was 16 min and 10 sec, where each repetition lasted for 3 min and 14 sec. The images were plotted with Python and adjusted by applying a 3 × 3 kernel with values of 0.1. The MR spectra of TEG-water evaporation, shown in Fig. 3e, were recorded with the ISIS pulse sequence by applying a 3000 ms repetition time, 8 scans (64 ISIS scans), on a 5 × 5 × 5 mm voxel, and a 5 Hz exponential line broadening. The scan time of each spectrum was 31 sec.

## Data availability
The authors declare that the data supporting the findings of this study are available within the paper and its Supplementary Information files. Should any raw data files be needed in another format they are available from the corresponding author upon request.

## Code availability
The Python code used for adjusting the operating frequency of the transducers is available on https://git.chalmers.se/bordes/OmniLev.

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

## Acknowledgements

The authors are grateful for the financial support from the Swedish Research Council (VR), Public, Sweden, (2018-04196) granted to L.E. and R.B., and the Swedish Foundation for Strategic Research (SSF), Non-Profit, Sweden, (ITM17-0436), granted to R.B. We would also like to thank Dr. Pavel Iouchmanov for the valuable discussions, and Professor Sergey Dvinskikh, from the Division of Applied Physical Chemistry, at KTH Royal Institute of Technology for carrying out the reference $^1$H-NMR spectrum of canola oil on the 300 MHz magnet at their facility.

## Author contributions

R.B. and L.E. conceptualized the idea and collected funding. R.B. designed and built the demagnetized acoustic levitator and the mechanical lift. S.M.A. and L.S. contributed equally. S.M.A., L.S., and F.G. conducted the experiments. In addition, S.M.A. and L.S. developed the methodology, performed data analysis, and drafted the manuscript. S.M.A. performed the acoustic pressure simulations, MR image analysis, and formatted the Figures. L.S. performed the theoretical analysis and explanation related to the droplet shape and chemical shift. R.B., L.E., and D.B. supervised and provided input. All authors reviewed, edited, and approved the final manuscript.

## Funding

## Competing interests

The authors declare no competing interests.
