## [Transparent Peer Review file · Nature Communications]

Contact-free magnetic resonance imaging and spectroscopy with acoustic levitation

Corresponding Author: Dr Romain Bordes

Version 0:

Reviewer comments:

Reviewer #1

(Remarks to the Author)

The submitted manuscript presents an experimental study investigating an innovative approach to contact-free magnetic resonance studies using acoustic levitation, which holds significant potential for observing real-time dynamic phenomena at the molecular level. The detailed experimental methodology and promising results are commendable. However, a deeper comparative analysis with traditional methods and a broader discussion on the interpretation and practical applications of the data would further enhance the study's impact and significance. Certain revisions could further improve its clarity and relevance, as discussed below:

1. The manuscript lacks detailed physical insight. While the combination of acoustically levitated samples with magnetic diagnostics is promising, a more thorough exploration of the underlying physical mechanisms is necessary.
2. Fig. 1d-f: Although the volume change due to evaporation is acknowledged, the temporal resolution of the current method is questionable. Can the volume change be validated using an alternative method?
3. For the benefit of the broader readers, the principles of acoustic levitation and the physicochemical properties of the samples used in this study should be included.
4. The plateau in water evaporation observed between 6-15 minutes needs clarification. Previous studies suggest continuous evaporation due to diffusion. The interaction between water droplets and acoustic fields plays a critical role in evaporation dynamics and should be discussed in more detail.
5. An uncertainty analysis should be added to the revised manuscript. The methodology for quantifying droplet volume and aspect ratio from the blurred images in Fig. 1 should be explained. Additionally, the authors should address the reasons for the differences in deposited volumes between samples.

Minor Improvements:

1. An "Introduction" heading should be added before the first paragraph of the main text.
2. Fig. 1: The label "f)" is missing before "coronal...".
3. Figs. 1d-f and 3a-c: An explanation for the color bar should be added.

Reviewer #2

(Remarks to the Author)

"Contact-free Magnetic Resonance Imaging and spectroscopy with acoustic levitation" by S-M Argyri et al.
The manuscript presents an application of acoustic levitation for a contact-free MRI and NMR of droplets. Effects of droplet shapes on the chemical shift, as well as the droplet evaporation, were examined. MR images of the levitating drops were acquired to estimate droplets' parameters and evolution. Several experimental challenges had to be resolved along the way (the RF energy deposition leading to the droplets bursting was one of the more intriguing ones). The findings can lead to further research in contact-free NMR studies of physical and chemical transformations of levitating fluids. The paper reports

interesting and well-organized research; it is also clearly written. I recommend it for publishing.

Minor comments:

The strength and duration of spoilers were indicated as important as they affect the signal quality. I assume they were gradient spoiler as they are later mentioned as "post-acquisition gradient pulses". However, I couldn't find their parameter, only that the "minimum spoiler length and duration" were applied. Can you please clarify? Also, it's unclear how their application would lead to a potential temperature increase. Is the temperature concern related to a potential need for greater number of scans/RF pulses applied, or is it something to do with the gradient heating (unlikely)?

It would be useful to learn more on the limitations on the droplet size, both from the NMR perspective (volumes used; RF probe requirements), and from the acoustic perspective (how scalable the setup is). After the stability limits are exceeded, the droplet dynamics should change considerably, potentially leading to research interest on its own.

Was temperature control of the droplet contents (e.g., with an IR camera) considered, given the effects of RF deposition? The temperature changes during, and after RF pulses would result in changes to viscosity and surface tension of the fluid.

I disagree with the last sentence in Introduction/Main: "Dynamic processes and phenomena can be studied on a molecular level without the intervention of surface boundaries or surface-induced effects". The levitation is achieved by means of a strong acoustic wave in air acting on the fluid – through its surface. An increased acoustic pressure can lead to the loss of sphericity. It can also induce acoustic streaming/microcirculation inside the droplet (see, for example, Saha et al, Phys Lett A, 2012, 376). In fact, measuring these non-linear effects with NMR/MRI is another potential application of the reported research. To sum it up, the contribution of surface-induced effects on the processes inside the acoustically levitating droplet can be considerable.

Reviewer #3

(Remarks to the Author)

The authors describe measurements of acoustically levitated droplets by magnetic resonance imaging, and spectroscopy using an compact multiple-transducers acoustic levitator, customised by the authors for better stability of levitated samples and capability to put the levitator in the high magnetic field. Chemical shifts in magnetic resonance spectra due to the change of the aspect ratio of the droplets reported. Time resolved study of evaporated solution shows clear transition of signals.

The achievement of the device development, proof of concept by the results of measurements is remarkable. However, I still believe authors need some revisions of the manuscript. I wonder if the device is useful for practical measurements such as photo chemical reaction, enzyme reaction etc. Authors might think of completely different scientific applications from my understandings. It would be better to describe the future perspective, what kind of applications in mind and possibly mention to a practical application in the manuscript.

comments:

- What's the exact volume of hexadecane droplets ? from the MR imaging, looks like few microliters. Without this information, I cannot judge if the results of imaging is reasonable.
- Positional stability should be important for imaging. In Fig 6 of ref 18, average displacement is described but in much shorter time resolution, how stable are the droplets ? Higher pressure induces more instability with levitators in general.
- Wouldn't it be possible to give feedback control of the acoustic pressure to stabilise the aspect ratio ? Obviously, one should avoid chemical shifts due to the aspect ratio change for practical use.
- Acoustic streaming is not mentioned to the manuscript. Doesn't it affect to the results of imaging/spectroscopy ? Lower pressure for better stability, more acoustic streaming occurs.
- What are the time resolution of the measurements ? How long does it take to collect an image of FLASH, True-FISP and RARE or to measure a spectrum of ISIS MR spectroscopy ? This information defines for what kind of time resolved measurement can be achievable with this device.

Version 1:

Reviewer comments:

Reviewer #1

(Remarks to the Author)

The authors have responded to all of the reviewer's comments and suggestions. I recommend the manuscript for publication.

Reviewer #2

(Remarks to the Author)

I am satisfied with the changes to the manuscript in response to both my and other reviewers' comments. In my opinion, the paper can now proceed to the publication stage.

Reviewer #3

(Remarks to the Author)

The authors revised their manuscript and added important numbers for the experiments to evaluate if the setup is useful for the applications of readers. It is much clearer that the acoustic levitator can be used as a versatile, contact-free sample container without concerns of temperature changes, aspect ratio due to the acoustic pressure. I believe it is worth to publish their manuscript to attract readers to use this instruments for their time-resolved experiments.

I have some minor comments and questions for your answers for my previous comments.

comment 1, volume of the droplets

- What's the reason not to use micro pipettes to load the same amount of droplets ? In addition, why would you use bigger amount of droplets to evaluate the evaporation ? Smaller amount of droplets shows faster evaporation. would be better to observe till the droplets completely evaporate.

comment 4, acoustic streaming

- Did i understand correctly that phased array levitators cause less streaming than single axis ones ? If so would be nice to mention to this fact in this paper. If others try to do mixing/soaking experiments with phased array ones, it would be difficult.

Response to reviewers' comments

We genuinely thank the reviewers for their feedback. We have received it in the spirit of promoting the quality of the Journal and the advancement of the field. Below, we provide answers to the points raised by the reviewers. We have revised certain sections of the manuscript in accordance with the comments and suggestions.

Reviewer #1

The submitted manuscript presents an experimental study investigating an innovative approach to contact-free magnetic resonance studies using acoustic levitation, which holds significant potential for observing real-time dynamic phenomena at the molecular level. The detailed experimental methodology and promising results are commendable. However, a deeper comparative analysis with traditional methods and a broader discussion on the interpretation and practical applications of the data would further enhance the study's impact and significance. Certain revisions could further improve its clarity and relevance, as discussed below:

Comment 1:

The manuscript lacks detailed physical insight. While the combination of acoustically levitated samples with magnetic diagnostics is promising, a more thorough exploration of the underlying physical mechanisms is necessary.

Response

Thank you for this comment. However, it is unclear to us, which physical mechanism the reviewer is referring to. The physics regarding magnetic resonance and acoustic levitation have been extensively reported individually elsewhere [RX1-RX8].

Herein, we introduced a demagnetized acoustic levitator (*i.e.*, modified so as to not contain any parts sensitive to a magnetic field), in the MR magnet with the aim of studying self-standing droplets. We explored the influence of the magnetic field on the performance of the transducers of the levitator and we showed that it was possible to levitate a microliter droplet and acquire MR images and spectra. We would like to point out that the two fields (magnetic and acoustic) do not have any direct interactions. However, in case strong RF pulses are used, it is possible to warm up the medium (*i.e.*, air) which will change the speed of sound and consequently influence the acoustic pressure field. It is possible to simulate this based on Ref. [RX9]. From our experience these temperature differences would not be greater than 2 °C; but below we simulated and compare the acoustic pressure along the x-, y- and z- axis, at 20 °C, 25 °C, and 30 °C. We can see that there is practically no change in acoustic pressure near the acoustic node. So, we can safely state that operating an acoustic levitator when major magnetic pulse sequences are applied is possible and that the heating effect is negligible. We have now added more information towards that direction in lines 14-20 and lines 68 together with the Figure we present below in SI.

Figure 1: Normalized acoustic pressure of levitator Mk3 at 20 (blue), 25 (orange), and 30 °C (red) along the a) z, b) x, and c) y axis.

Furthermore, in our study we confirmed that the shape of the levitating droplet affects the local magnetic field inside the droplet, resulting in a chemical shift, which has been extensively described in the Methods.

We hope that these points cover the underlying mechanisms the reviewer is referring to.

[RX1] Slichter, Charles P. *Principles of magnetic resonance*. Vol. 1. Springer Science & Business Media, 2013.

[RX2] Vlaardingerbroek, Marinus T., and Jacques A. Boer. *Magnetic resonance imaging: theory and practice*. Springer Science & Business Media, 2013.

[RX3] Chizhik, Vladimir I., et al. *Magnetic resonance and its applications*. No. 15506. Cham: Springer International Publishing, 2014.

[RX4] Andrew, Edward Raymond. "Nuclear magnetic resonance." *Nuclear Magnetic Resonance* (2009).

[RX5] Yarin, A. L., M. Pfaffenlehner, and C. Tropea. "On the acoustic levitation of droplets." *Journal of Fluid Mechanics* 356 (1998): 65-91.

[RX6] Andrade, Marco AB, Nicolás Pérez, and Julio C. Adamowski. "Review of progress in acoustic levitation." *Brazilian Journal of Physics* 48 (2018): 190-213.

[RX7] Andrade, Marco AB, Asier Marzo, and Julio C. Adamowski. "Acoustic levitation in mid-air: Recent advances, challenges, and future perspectives." *Applied Physics Letters* 116.25 (2020).

[RX8] Zang, Duyang. "Acoustic Levitation." *From Physics to Applications* (2020).

[RX9] levitate, <https://github.com/AppliedAcousticsChalmers/levitate> (2021).

Comment 2:

Fig. 1d-f: Although the volume change due to evaporation is acknowledged, the temporal resolution of the current method is questionable. Can the volume change be validated using an alternative method?

Response

Thank you for this comment. We do agree that there is a challenge here in terms of spatial and temporal resolution.

Since there is no visual access to the levitating object while it is inside the MR bore, the only other option to follow the volume evolution, besides MRI, is through magnetic resonance spectroscopy (MRS). Another option, for volatile droplet as in the case of a droplet of water is levitating, we can

evaluate the volume by acquiring a series of spectra and integrating the signal that corresponds to the -OH protons of water. What we will observe is a continuous decrease of the signal intensity (*i.e.*, spectral integral of the peak), which is directly proportional to the volume of the droplet. However, this is applicable only for volatile solvents and not in the case where a non-volatile or a mixture of volatile and non-volatile substances is studied (as in the case of the TEG/water system). Furthermore, at least one magnetic resonance image in the beginning of the experiment would be required to verify the presence of a levitating droplet and the starting volume. Another option is to increase the image resolution of the MRI as it was achieved with the pulse sequence True-FISP (Fig. S11, Supporting Information) but with less artifacts.

In the current paper, we have validated outside the magnet the volumes of the droplets using a camera and image processing, as in Ref [10]. However, the kinetics of evaporation are significantly different. As a matter of fact, the pulse sequence has a warming up effect on the volatile droplet that promotes evaporation. We have added a sentence on that line 68.

Furthermore the volume calculated from the MRI data is in agreement with our previous publications using similar levitators. This is clarified line XX

[Ref 10] Argyri, S.-M., Evenäs, L. & Bordes, R. Contact-free measurement of surface tension on single droplet using machine learning and acoustic levitation. *J. Colloid Interface Sci.* 640, 637–646 (2023).

Comment 3:

For the benefit of the broader readers, the principles of acoustic levitation and the physicochemical properties of the samples used in this study should be included.

Response

We have given a short description of the acoustic pressure field lines 13-16; we have added a clearer description of acoustic levitation principle and added more references in the same place.

Comment 4: The plateau in water evaporation observed between 6-15 minutes needs clarification. Previous studies suggest continuous evaporation due to diffusion. The interaction between water droplets and acoustic fields plays a critical role in evaporation dynamics and should be discussed in more detail.

Response:

It is unclear which publications the reviewer is referring to. We report two experiments regarding evaporation; one with a TEG/water mixture (50 wt%), which can inevitably lead to a plateau when all water is evaporated and one with only water for which the signal completely disappears. For the latter we did not plot the intensity as function of time as it was not giving more insights.

For the 50 wt% aqueous TEG droplet, with volume of approximately 2.5 μL , a plateau was observed after the droplet evaporated for 6 minutes under acoustic levitation. This observation was made on different droplets through imaging by following the pixel intensity over time, and spectroscopy, by following the integral of the -OH signal and the same plateau was reached. The fact that there is no significant amount of water in the droplet after 6 min was confirmed by the ratio between the -OH and main chain-TEG protons which is 1:6 (2 -OH and 12 H in the main TEG chain: CH_2) (see line 176-177). Furthermore, we did the same experiments outside the magnet and got the same observation, *i.e.* a plateau, though the kinetics were different.

The evaporation dynamics of acoustically levitated droplets have been reported in previous studies [RX11]. Studies have shown that spherical droplets follow the d^2 law of evaporation. This law does not account for the presence of an acoustic pressure field but determines the theoretical change of volume of a spherical object. Other studies have mentioned the influence of humidity and added flow on the evaporation rate; however, this is also not related to the acoustic pressure field but to the environmental conditions.

Furthermore, due to the high-density difference between the medium, Z_1 (*i.e.*, air) and the droplet, Z_2 (e.g., water+TEG) 99.9% of the acoustic waves will reflect on the surface of the droplet and will not influence the droplet itself, due to acoustic impedance. This property relates to the density difference between the medium and the droplet. Where the larger the density difference more waves will be reflected on the droplet's surface.

To calculate the percentage of waves reflected we can use the following equation:

$$a = \left(\frac{Z_2 - Z_1}{Z_1 + Z_2} \right)^2$$

Where, the acoustic impedance of air is $Z_1=420$ kg/s and for water it is $Z_2= 1.48*10^6$ kg/s.

So, $\alpha = 0.999425$. In the case of TEG, the density of the droplet will increase so the percentage of the waves reflected will be higher too.

So, we can safely state that the acoustic pressure field only plays the role of keeping the droplet aloft.

Overall, the objective of the specific experiment was to determine the potential of MRI and MRS on a single, microliter droplet. And it was found that we could not only follow the evaporation accurately but also derive information regarding the molecular state of the system in solution. It is clearly stated line 183-184.

[RX11] Zang, Duyang. "Acoustic Levitation." *From Physics to Applications* (2020).

Comment 5:

An uncertainty analysis should be added to the revised manuscript. The methodology for quantifying droplet volume and aspect ratio from the blurred images in Fig. 1 should be explained. Additionally, the authors should address the reasons for the differences in deposited volumes between samples.

Response:

An analysis on how the volume was calculated is already present in the Supporting Information (see Fig. S10, and S13). We apologized for the lack of error bars in Fig. 1h, and Fig. 1i. They have now been added.

The droplets are deposited by hand with a plastic or glass syringe equipped with a metallic needle (not dispensed with a micropipette), so the initial volume of the sample can vary. Depending on the operating voltage of the levitator (*i.e.* the acoustic pressure), and the surface tension of the liquids studied there is a range of volumes that may be deposited (see Ref. 10 in the manuscript). The objective in this publication was not to repeatedly levitate the same volume of droplets but to show the possibilities of this technique. In the case where the initial volume is important for the repeatability of the experiments the use of a dispensing device is recommended.

[Ref 10] Argyri, S.-M., Evenäs, L. & Bordes, R. Contact-free measurement of surface tension on single droplet using machine learning and acoustic levitation. *J. Colloid Interface Sci.* 640, 637–646 (2023).

Minor Improvements:

1. An "Introduction" heading should be added before the first paragraph of the main text.

This has now been fixed line 12. Thank you.

2. Fig. 1: The label "f)" is missing before “coronal...”.

This has now also been fixed in the legend of Fig. 1. Thank you.

3. Figs. 1d-f and 3a-c: An explanation for the color bar should be added.

This has now also been fixed in the legend of Fig. 1. Thank you.

Reviewer #2

The manuscript presents an application of acoustic levitation for a contact-free MRI and NMR of droplets. Effects of droplet shapes on the chemical shift, as well as the droplet evaporation, were examined. MR images of the levitating drops were acquired to estimate droplets' parameters and evolution. Several experimental challenges had to be resolved along the way (the RF energy deposition leading to the droplets bursting was one of the more intriguing ones). The findings can lead to further research in contact-free NMR studies of physical and chemical transformations of levitating fluids. The paper reports interesting and well-organized research; it is also clearly written. I recommend it for publishing.

Comment 1:

The strength and duration of spoilers were indicated as important as they affect the signal quality. I assume they were gradient spoiler as they are later mentioned as “post-acquisition gradient pulses”. However, I couldn't find their parameter, only that the “minimum spoiler length and duration” were applied. Can you please clarify? Also, it's unclear how their application would lead to a potential temperature increase. Is the temperature concern related to a potential need for greater number of scans/RF pulses applied, or is it something to do with the gradient heating (unlikely)?

Response:

Thank you for your comments. In the software that is operating the instrument, it is possible to minimize the values of the spoiler strength and duration by setting them to zero. Depending on the pulse sequence and the input parameters, different minimal values were automatically assigned. Since this parameter is adjusted automatically for each measurement, we did not mention all the values in the manuscript. To clarify this, we have added a sentence in line 257.

Thank you for your comment on the increase of temperature. It was a typo in the manuscript and has now been fixed (line 128). The spoilers themselves do not increase the temperature of the sample, as their input energy is small in comparison to the RF pulses. We have noticed experimentally that a slightly faster evaporation takes place during the MR experiments. This was related to the input energy of the pulse sequences, which can even lead to the droplet bursting as shown in Fig. S9, Supporting Information. However, we did not quantify exactly the difference as many parameters (e.g. pulse sequence, size of the droplet, humidity, etc.) can affect the outcome.

Throughout the manuscript “spoiler” has been replaced by “gradient spoiler”.

Comment 2:

It would be useful to learn more on the limitations on the droplet size, both from the NMR perspective (volumes used; RF probe requirements), and from the acoustic perspective (how scalable the setup is). After the stability limits are exceeded, the droplet dynamics should change considerably, potentially leading to research interest on its own.

Response:

This is an important comment as it underscores the possibilities of our setup. The size of the droplet depends largely on the wavelength of the ultrasonic transducers followed by the driving voltage and surface tension of the levitated liquid. In this case, the ultrasonic transducers were operated at their resonant frequency (40 kHz). Consequently, the wavelength of the acoustic waves is 8.575 mm (for a speed of sound of 343 m/s). This means that the size of the acoustic node is 4.2875 mm, which corresponds to the upper limit of the diameter of the levitating sample (*i.e.*, volume of 41.3 μL for a spherical droplet). However, this value is never achieved and in most cases droplet above 10 μL would burst. This is because of the capillary length of the liquid, parameter below which the surface tension governs the shape of the droplet. Above the capillary length, the gravitational forces cause the droplet to flatten under its own weight and eventually form two droplets. In the case of water, the capillary length is 2.718 mm, meaning that the maximum volume for a spherical droplet would be 10.5 μL .

In line 16 we stated a realistic volume range of the levitating samples (0.5 – 5 μL) based on empirical and practical observations. Furthermore, in the section “acoustic levitation” we have added a short paragraph in lines 212-214 to indicate the boundaries in terms of volume and density of the acoustic levitation capacities of the setup, and added a sentence in the conclusion, in line 190.

Comment 3:

Was temperature control of the droplet contents (e.g., with an IR camera) considered, given the effects of RF deposition? The temperature changes during, and after RF pulses would result in changes to viscosity and surface tension of the fluid.

Response:

The temperature of the droplets was not monitored during the magnetic resonance measurements with an IR camera, mainly because it was not practical to insert such an instrument in the bore/magnetic field. However, through a continuous acquisition of MR spectra on non-volatile droplets, as in the case of hexadecane, it is visible that for a constant voltage the chemical shift of the signal was affected less than a 0.05 ppm over a period of 30 minutes while acquiring an MR spectrum with the ISIS pulse sequence, every 30 seconds (Fig. S17, Supporting Information). In case where the temperature would be affected, a more pronounced change in the chemical shift would be recorded. This has now been pointed out in the manuscript, in lines 138-139.

Comment 4:

I disagree with the last sentence in Introduction/Main: “Dynamic processes and phenomena can be studied on a molecular level without the intervention of surface boundaries or surface-induced effects”. The levitation is achieved by means of a strong acoustic wave in air acting on the fluid – through its surface. An increased acoustic pressure can lead to the loss of sphericity. It can also induce acoustic

streaming/microcirculation inside the droplet (see, from example, Saha et al, Phys Lett A, 2012, 376). In fact, measuring these non-linear effects with NMR/MRI is another potential application of the reported research. To sum it up, the contribution of surface-induced effects on the processes inside the acoustically levitating droplet can be considerable.

Response:

Thank you for your comment and apologies for the lack of clarity. In that sentence we wanted to point out that the levitating droplet is only surrounded by the medium (*i.e.*, air) and not contained within solid surfaces, as in the case of sample containers, or deposited on a solid surface, which in the case of *e.g.*, evaporation influences the dynamics of the phenomenon (which, as stated in the manuscript, is important for applications *e.g.*, related to drying of coatings, or spraying application). For the sake of clarity, we have modified the sentence as follows: “Dynamic processes and phenomena can be studied on a molecular level without the presence of a container”, line 48.

The acoustic levitation takes place due to the presence of a stable acoustic pressure field with alternating areas of low (*i.e.*, acoustic nodes) and high (*i.e.*, acoustic anti-nodes) pressure, as in the case of standing waves. Acoustic streaming refers to the steady flow of liquid induced by the absorption of sound waves. This (may) occur close to the surface of the levitating sample depending mainly on the density of the sample and the applied acoustic pressure. As a result, acoustic streaming is an effect related to the acoustic levitation of the object but it is not what induces the levitation. Surface induced effects originating from the acoustic pressure field could be introduced by actively altering the acoustic pressure (*i.e.*, driving voltage) at a very high rate (*e.g.* frequency), causing surface oscillations. Furthermore, using a very small voxel size (potentially $\leq 0.1 \times 0.1 \times 0.1$ mm) would be necessary. In this study we applied constant acoustic pressure throughout the measurements and a voxel size of $\geq 3 \times 3 \times 3$ mm, which is at least 2 times larger than the diameter of the droplets. In general, no such effects have been experimentally observed outside (through high-speed cameras) or inside the magnet under constant acoustic pressure.

Reviewer #3

The authors describe measurements of acoustically levitated droplets by magnetic resonance imaging, and spectroscopy using an compact multiple-transducers acoustic levitator, customised by the authors for better stability of levitated samples and capability to put the levitator in the high magnetic field. Chemical shifts in magnetic resonance spectra due to the change of the aspect ratio of the droplets reported. Time resolved study of evaporated solution shows clear transition of signals.

The achievement of the device development, proof of concept by the results of measurements is remarkable. However, i still believe authors need some revisions of the manuscript. I wonder if the device is useful for practical measurements such as photo chemical reaction, enzyme reaction etc. Authors might think of completely different scientific applications form my understandings. It would be better to describe the future perspective, what kind of applications in mind and possibly mention to a practical application in the manuscript.

Response:

Thank you for your supportive comments. Photochemical and enzymatic reactions [X1-X2], protein crystallization [X3] and other studies [X4] have already been performed with acoustic levitation (though not combined with NMR, yet). As suggested, we have added that lines 201 to point out more clearly the possibilities of our set-up.

[X1] Tobon, Yeny A., et al. "Photochemistry of single particles using acoustic levitation coupled with Raman microspectrometry." *Journal of Raman Spectroscopy* 48.8 (2017): 1135-1137.

[X2] Weis, David D., and Jonathan D. Nardoizzi. "Enzyme kinetics in acoustically levitated droplets of supercooled water: A novel approach to cryoenzymology." *Analytical chemistry* 77.8 (2005): 2558-2563.

[X3] Tsujino, Soichiro, and Takashi Tomizaki. "Ultrasonic acoustic levitation for fast frame rate X-ray protein crystallography at room temperature." *Scientific reports* 6.1 (2016): 25558.

[X4] van Wasen, Sebastian, et al. "Miniaturized Protein Digestion Using Acoustic Levitation with Online High Resolution Mass Spectrometry." *Analytical Chemistry* 95.8 (2023): 4190-4195.

[X5] Pierre, Zakiah N. *Acoustically-levitated drop reactor (LDR) employable for kinetics measurements of biochemical networks*. Diss. University of Illinois at Urbana-Champaign, 2011.

[X6] Lorenzen, Elke, and Geoffrey Lee. "Slow motion picture of protein inactivation during single-droplet drying: A study of inactivation kinetics of l-glutamate dehydrogenase dried in an acoustic levitator." *Journal of pharmaceutical sciences* 101.6 (2012): 2239-2249.

Comment 1:

What's the exact volume of hexadecane droplets? from the MR imaging, looks like few microliters. Without this information, i cannot judge if the results of imaging is reasonable.

Response:

Yes, as stated in the introduction lines 16 all droplets are in the microliter regime (0.5-5 μL). In the case of the droplet of hexadecane, the volume was estimated from the analysis shown in Fig. S13, and the results are reported in Table S2, in Supporting Information. The volume was found to be approximately 3.75 μL . This has now been added in the legend of Figure 1 and Figure 2.

Comment 2:

Positional stability should be important for imaging. In Fig 6 of ref 18, average displacement is described but in much shorter time resolution, how stable are the droplets ? Higher pressure induces more instability with levitators in general.

Response:

In Ref. 18 it was shown that for driving voltages in the range of 6.5 to 12 V, the average displacement was in the range of $\pm 10 \mu\text{m}$ outside of the magnet. The displacement was determined with imaging by extracting the contour of the droplets.

Here, such measurements with a camera were not possible inside the magnet. Yet, through the MRIs we collected, we can evaluate the spatial stability of the droplet. In the case of the evaporation of the water droplet, a maximum displacement of 0.5 mm was observed on rare occasions (Fig. S10). However, in the case of hexadecane, non-detectable displacement was found within the experimental limit of the measurements (Fig. S13), and within the timeframe of the data acquisition. Furthermore, the MR resolution is in all cases larger than 250 μm , which is 25 times larger than the measured displacement in Ref 18. This has now been clarified in the paper in lines 102.

[Ref. 18] Argyri, Smaragda-Maria, et al. "Customized and high-performing acoustic levitators for contact-free experiments." *Journal of Science: Advanced Materials and Devices* 9.3 (2024): 100720.

Comment 3:

Wouldn't it be possible to give feedback control of the acoustic pressure to stabilise the aspect ratio? Obviously, one should avoid chemical shifts due to the aspect ratio change for practical use.

Response:

This is a good point and in practice it could be done using imaging. In case a constant oblate shape is targeted, it should also be possible to develop a loop that would use the images in case changes would be recorded then a gradual increase of the driving voltage would be implemented until the initial shape is recovered. However, this method would be slow since the duration of an images acquisition is in the range of minutes.

However, the variation of chemical shift offers simpler option, and is faster. Having a feedback loop between chemical shift and driving voltage would enable a closed loop on the control of the shape. The minimum spectral acquisition time in our study was 30 sec, yet faster spectra acquisition should be possible with lower spectral resolution.

We believe that it is advantageous to follow potential chemical shift changes during the MR spectral acquisition, thus being aware of droplet shape changes, because it can provide qualitative insights into *e.g.*, change of surface tension during physicochemical reactions. In the case where the droplet shape needs to be kept constant, one could apply the lowest possible voltage, which will keep the droplet spherical (assuming that potential changes in the droplet are caused only by evaporation) throughout the experiment [X1].

[X1] Argyri, Smaragda-Maria, et al. "CO₂ induced phase transition on a self-standing droplet studied by X-ray scattering and magnetic resonance." *Journal of Colloid and Interface Science* (2024).

Comment 4:

Acoustic streaming is not mentioned to the manuscript. Doesn't it affect to the results of imaging/spectroscopy? Lower pressure for better stability, more acoustic streaming occurs.

Response:

In this study, we have not encountered any issues typically associated with acoustic streaming, *e.g.*, internal flow.

Previous reports have shown that acoustic streaming is quite low, with such multi-transducers acoustic levitators such as TinyLev [X9], as micron-sized particles sedimented within the droplet under levitation [X10-X11]. In the presence of acoustic streaming, it would not be the case, and the particles would be continuously agitated.

[X9] Marzo, Asier, Adrian Barnes, and Bruce W. Drinkwater. "TinyLev: A multi-emitter single-axis acoustic levitator." *Review of Scientific Instruments* 88.8 (2017).

[X10] Zeng, Hao, et al. "On evaporation dynamics of an acoustically levitated multicomponent droplet: Evaporation-triggered phase transition and freezing." *Journal of Colloid and Interface Science* 648 (2023): 736-744.

[X11] Bunio, Lyndon B., et al. "Evaporation and crystallization of NaCl-water droplets suspended in air by acoustic levitation." *Chemical Engineering Science* 251 (2022): 117441.

Comment 5:

What are the time resolution of the measurements ? How long does it take to collect an image of FLASH, True-FISP and RARE or to measure a spectrum of ISIS MR spectroscopy ? This information defines for what kind of time resolved measurement can be achievable with this device.

Response:

Depending on the pulse sequence and image resolution different, the acquisition time to acquire MRI was between 30s to 500s, with True-FISP being the fastest. The time intervals shown in Fig. 1 and Fig. 3 (see insets and legends), in the Methods (lines: 228-229, 232) and in the Supporting Information (Fig. S9, S10, and S11), are giving an idea of the acquisition time per image. All total acquisition times have now been clearly stated in the Methods.

REVIEWERS' COMMENTS

We would like to thank the reviewer for the additional comments. Please find below our answers to the comments regarding the experimental decisions and acoustic streaming.

We have also adjusted the format based on the requirements of the Journal and corrected some typos found during the revision.

Reviewer #1 (Remarks to the Author):

The authors have responded to all of the reviewer's comments and suggestions. I recommend the manuscript for publication.

Thank you for this positive evaluation.

Reviewer #2 (Remarks to the Author):

I am satisfied with the changes to the manuscript in response to both my and other reviewers' comments. In my opinion, the paper can now proceed to the publication stage.

Thank you for this positive evaluation.

Reviewer #3 (Remarks to the Author):

The authors revised their manuscript and added important numbers for the experiments to evaluate if the setup is useful for the applications of readers. It is much clearer that the acoustic levitator can be used as a versatile, contact-free sample container without concerns of temperature changes, aspect ratio due to the acoustic pressure. I believe it is worth to publish their manuscript to attract readers to use this instruments for their time-resolved experiments.

I have some minor comments and questions for your answers for my previous comments.

comment 1, volume of the droplets

- What's the reason not to use micro pipettes to load the same amount of droplets ? In addition, why would you use bigger amount of droplets to evaluate the evaporation ? Smaller amount of droplets shows faster evaporation. would be better to observe till the droplets completely evaporate.

Thank you for the comments. We choose to use a needle for practical reasons. The outer diameter of the needle is approximately 0.1-0.5 mm, while a plastic pipette tip has a larger diameter. As a result, the needle and the droplet have a small area of contact during deposition, which allows the droplet to detach easier from the needle. This allows us to have better control over the volume of the droplet; this is particularly important for liquids with high surface tension, such as water, which may be more difficult to detach from a pipette tip. Furthermore, to deposit the droplet in the acoustic node, typically one needs to position the syringe with the needle horizontally. Of course, for different applications, different methods of droplet injection could be more appropriate. We have actually used micropipettes when the levitator is placed horizontally on the stage of a Raman microscope [R1]

The main reason why large water droplets were used was to ensure that we could acquire a clear NMR signal over a long period of time, and to be able to collect as many good magnetic resonance

images as possible. As seen in Fig. S12 for example, we were able to collect 3 MRIs for the specific water droplet and pulse sequence parameters. On the 4th MRI the droplet had indeed completely evaporated.

comment 4, acoustic streaming

- Did i understand correctly that phased array levitators cause less streaming than single axis ones ? If so would be nice to mention to this fact in this paper. If others try to do mixing/soaking experiments with phased array ones, it would be difficult.

Based on our experience with multiple-transducers acoustic levitators, no internal flow has been observed. We have investigated the potential presence of internal flow for another manuscript which was recently published [R1]. Specifically, we levitated a droplet of water containing Al_2O_3 particles (average diameter of 180 μm) and silica particles (average diameter of 10 μm). In both cases we observed that the particles sediment at the bottom of the droplet, and no internal flow was observed (Fig. R1, Fig. R2). These results are in line with previous publications where precipitation of NaCl salt was observed within the droplet [R2]. We also levitated a water droplet with smaller particles ($< 1 \mu\text{m}$) but in that case, the colloidal suspension was stable over a period of time longer than the time needed for complete evaporation. Therefore, our results do not support the claims of internal flow. Interestingly, is it possible to add 2 miscible droplets together and mixing will take place due to the diffusion of the molecules, which would be very interesting to study with NMR. A parameter, related to the acoustic pressure field, that may potentially affect the mixing is the rotation of the droplet(s) around the central z-axis. In case one is interested in studying the diffusion of molecules, the rotation of the droplet should be controlled. There are already a few publications working towards that [R3-R4]. We have now added a few sentences about this in line 78-92.

Fig. R1: Water droplet containing Al_2O_3 particles that have sedimented at the bottom within 10 seconds.

Fig. R2: Water droplet containing SiO₂ particles that have sedimented at the bottom a) within 1 min, and b) after 10 min of water evaporation.

[R1] Argyri, Smaragda-Maria, et al. "Crystallization at the hexadecane/water interface observed under acoustic levitation." *Journal of Environmental Sciences* (2025).

[R2] Bunio, Lyndon B., et al. "Evaporation and crystallization of NaCl-water droplets suspended in air by acoustic levitation." *Chemical Engineering Science* 251 (2022): 117441.

[R3] Marzo, Asier, et al. "Holographic acoustic elements for manipulation of levitated objects." *Nature communications* 6.1 (2015): 8661.

[R4] Contreras, Victor, and Karen Volke-Sepúlveda. "Enhanced standing-wave acoustic levitation using high-order transverse modes in phased array ultrasonic cavities." *Ultrasonics* 138 (2024): 107230.